# Approaches to Targeting Bacterial Biofilms in Cystic Fibrosis Airways

**DOI:** 10.3390/ijms22042155

**Published:** 2021-02-22

**Authors:** Isaac Martin, Valerie Waters, Hartmut Grasemann

**Affiliations:** 1Division of Respiratory Medicine, Department of Paediatrics, The Hospital for Sick Children, University of Toronto, Toronto, ON M5G 1X8, Canada; hartmut.grasemann@sickkids.ca; 2Division of Infectious Diseases, Department of Paediatrics, The Hospital for Sick Children, University of Toronto, Toronto, ON M5G 1X8, Canada; valerie.waters@sickkids.ca; 3Department of Paediatrics and Translational Medicine, Research Institute, The Hospital for Sick Children, University of Toronto, Toronto, ON M5G 0A4, Canada

**Keywords:** *Pseudomonas aeruginosa*, biofilm matrix, anti-biofilm, exopolysaccharides, adjunctive therapies, antimicrobial resistance

## Abstract

The treatment of lung infection in the context of cystic fibrosis (CF) is limited by a biofilm mode of growth of pathogenic organisms. When compared to planktonically grown bacteria, bacterial biofilms can survive extremely high levels of antimicrobials. Within the lung, bacterial biofilms are aggregates of microorganisms suspended in a matrix of self-secreted proteins within the sputum. These structures offer both physical protection from antibiotics as well as a heterogeneous population of metabolically and phenotypically distinct bacteria. The bacteria themselves and the components of the extracellular matrix, in addition to the signaling pathways that direct their behaviour, are all potential targets for therapeutic intervention discussed in this review. This review touches on the successes and failures of current anti-biofilm strategies, before looking at emerging therapies and the mechanisms by which it is hoped they will overcome current limitations.

## 1. Introduction

Cystic fibrosis (CF) is a life limiting disease caused by mutations in the *cystic fibrosis transmembrane conductance regulator* (CFTR) gene. CF lung disease is characterized by inflammation as well as chronic bacterial infection, where bacteria become host-adapted pathogens. In CF, the lives of affected individuals are punctuated by pulmonary exacerbations in which bacterial driven inflammatory changes can lead to permanent scarring and decline in lung function [1,2]. *Pseudomonas aeruginosa* and *Staphylococcus aureus* are two most common pathogens in CF airways, although other organisms such as *Burkholderia* spp., non-tuberculous mycobacteria (NTM) spp., *Stenotrophomonas maltophilia*, *Achromobacter* spp., and fungi such as *Aspergillus* spp., have also been implicated in CF lung infection [3].

Understanding how bacteria survive in the lungs of patients despite years of antimicrobial therapy is important and may help designing new approaches to target chronic infections. Within the context of CF lung disease, bacterial antimicrobial resistance (AMR) increases over the lives of patients and limits the efficacy of antibiotics [4,5,6]. Higher rates of AMR, driven in large part by the selective pressures of antimicrobial treatments, are associated with decline in pulmonary function (as measured by forced expiratory volume in 1s, FEV1), as well as decreased quality of life outcomes and earlier mortality as disease progresses [7,8,9]. Resistance in the context of antimicrobial susceptibility testing (AST) involves measuring both innate properties of the bacteria like membrane permeability [10], as well as the expression of efflux pumps and the production of enzymes like β-lactamases that degrade antibiotics [11]. Conventional planktonic-based AST fails to take into account the biofilm lifestyle and resistance that result from social adaptation to the local environment. Although conventional AST is the laboratory cornerstone for antimicrobial selection in the treatment of acute infections, this method falls short of predicting clinical outcomes in CF [12,13].

Bacteria exhibit social behaviours and live in aggregates or communities of organisms suspended within a matrix of self-secreted proteins—termed a biofilm. The main constituents of this matrix are extracellular DNA, exopolysaccharides, lipids, proteins and metabolites [14]. These constituents surround the bacterial community, offering physical and electrostatic protection from antimicrobials, host defenses and other environmental stressors [15]. Channels for nutrients and metabolites running through the structure of protein scaffolding allow for differential access to nutrients, which creates a heterogeneity of bacterial populations with differing rates of metabolic activity and cell division [16]. The high level of population heterogeneity within the biofilm gives rise to ‘persister cells’, a metabolically dormant phenotype. Persister cells represent a small fraction of the bacterial population but are present in higher numbers within slow-growing biofilms compared to planktonically grown bacteria, perhaps due to differences in access to nutrients or as a population level survival mechanism. These sub populations of cells can survive high doses of bactericidal agents [17,18,19].

We now understand that the predominant mode of growth of bacteria in chronic infections is within biofilms rather than planktonically, which may be a transitional phase in the growth and spread of bacteria. Biofilms are a recognized hurdle to the successful treatment of chronic wound, ear and eye infections, as well as infections involving prosthetic medical devices [20]. Bacterial biofilms require a substratum to bind to, like that of a wound, or the surface of a foreign medical device, like a urinary catheter. However, in the lungs of individuals with CF, bacteria often aggregate to themselves, forming clusters of communities suspended within the airway mucus [21,22,23].

Each stage in the life cycle of a biofilm is summarized in Figure 1 from attachment, maturation, dispersal and colonization. The bacteria themselves and the components of the extracellular matrix, in addition to the signaling pathways that direct their behaviour, are all potential targets for therapeutic intervention. This review outlines the evidence for anti-biofilm agents currently in use and explores upcoming therapies and their targets. Recognizing the diversity of agents and targets, there is an emerging emphasis on the development of specific anti-biofilm therapies for adjuvant use with current antibiotics, for the exploitation of dual mechanism bacterial killing. 

## 2. Strategy: Direct Antibacterial Action

It has been shown that minimum biofilm eradication concentrations can be 100–1000 fold higher than the minimum inhibitory concentration (MIC) of antibiotics against planktonic grown bacteria [24]. Achieving these concentrations of antibiotics either systemically, through oral or intravenous administration, or via nebulization and inhalation, is difficult [25,26,27]. Thus, it becomes necessary to evaluate ways in which antibiotics can be effective as antibiofilm agents and where they fail, in order to optimize the efficacy of our treatments. Many antimicrobial agents are limited in action by the characteristics of bacterial biofilms. For instance, antibiotics including β-lactams, fluoroquinolones and aminoglycosides target actively dividing or metabolically active cells. Persister cells, cells with low metabolic activity embedded within the matrix, are refractory to treatment. Furthermore, components of the matrix itself provide a physical barrier to the dispersal and penetration of antimicrobials. *In vitro P. aeruginosa* work has demonstrated that the use of the aminoglycoside tobramycin—a mainstay in the treatment of Gram-negative infections—kills only the metabolically active layer of cells at the extremity of the biofilm, without the ability to penetrate the interior of the aggregate [28]. The polymyxin antibiotic colistimethate sodium (colistin), on the other hand, acts as a cationic detergent and binds to the negatively charged lipopolysaccharide present in the outer membrane of *P. aeruginosa* and has no such requirement for metabolic activity to take effect [28].

### 2.1. Dual Antibiotic Administration

The current practice of using antibiotic combinations that target different mechanisms of action to treat CF pulmonary infections may be more effective at killing bacterial biofilms than single antibiotics use. Colistin-tobramycin combinations demonstrated superior efficacy in comparison to monotherapy in the eradication of *P. aeruginosa* biofilms, both in vitro and in a rat lung infection model [28]. In addition, colistin-ceftolozane/tazobactam and colistin/meropenem combinations demonstrated both reduced bacterial counts of biofilm-embedded *P. aeruginosa*, and less colistin resistance when compared to colistin administration alone [29]. Biofilm models of both *Acinetobacter baumannii* and *P. aeruginosa* also showed that colistin-aminoglycoside combinations were efficacious in the eradication of persister cells otherwise unaffected by monotherapy [30,31].

### 2.2. Enhancing Antibiotic Efficacy and Delivery

One strategy to optimize antibiotic delivery to cells distributed within the biofilm is with liposomal preparations. In this approach, antibiotics are encapsulated in phospholipid vesicles that can fuse to cell membranes, allowing for improved drug delivery. Early studies showed that both charge [32] and size [33] were important factors in reaching bacterial cells within the biofilm. Imaging studies have confirmed penetration of liposomal amikacin particles deep into the layers of both non-tuberculous mycobacterial and *P. aeruginosa* biofilms [34,35]. Randomized trials of inhaled liposomal amikacin demonstrated non inferiority to tobramycin inhalation solution in *P. aeruginosa* lung infection in patients with CF [36,37]. More recently, a randomized trial using the same preparation for CF patients with NTM infection, resulted in culture conversion and improved performance on a 6 min-walk-test, although it did not meet the primary endpoint of reduction of sputum bacterial density [38]. The approach of developing liposomal formulations of antibiotics is the focus of other extensive reviews on the subject [37,39,40].

Antimicrobial peptides (AMPs) are a class of natural and synthetic proteins that vary greatly in size and function. They have been the focus of research looking into many diverse therapeutic applications and have shown promise both as antimicrobials in their own right, but also as adjuncts to standard antibiotics in combatting biofilm infections [41,42]. One such novel cationic AMP, 6K-F17, was seen to effectively kill biofilms and reduce biofilm biomass when used with tobramycin against multi-drug resistant CF *P. aeruginosa* isolates [43]. Still other AMPs have been shown to reduce LPS-induced pro-inflammatory responses [44,45]. The multitude of proposed functions—from direct antimicrobial action, to potentiation of antibiotics to the modulation of inflammatory responses known to be detrimental in infection—make AMPs an exciting adjunctive therapy in CF biofilm infections.

### 2.3. Bacteriophage

Bacteriophages are viruses that use bacteria as their hosts to complete their life cycle. There are different types of such viruses, including those that are lysogenic or filamentous. Lysogenic phages incorporate themselves into the host genome and can confer resistance genes [46] while filamentous phages are themselves a component of the biofilm matrix [47,48]. Recently, there has been a resurgence in interest of lytic phages used as an anti-biofilm agent. Lytic phages bind to, replicate within and lyse specific bacterial hosts and are present in the environment wherever there are bacteria. There are many proposed benefits to bacteriophage therapy—among them, the ability to replicate at sites with the highest burden of infection and the ability to target specific bacterial species, leaving the remaining host microbial community intact. In contrast to most conventional antibiotics that require actively dividing bacterial cells, bacteriophages have the ability to lyse metabolically dormant cells. A study investigating certain *S. aureus* phages demonstrated infection and lysis of metabolically dormant persister cells [49]. Some lytic phages specific to *P. aeruginosa* employ the use of exopolysaccharide degrading enzymes that can degrade biofilms, which may prove beneficial in chronic, mucoid infections. Early reports of *P. aeruginosa* lytic phages demonstrated expression of alginate lyase, an enzyme that degrades alginate (exopolysaccharide produced by many types of bacteria, but particularly *P. aeruginosa*), allowing the phages to penetrate into deeper layers of the bacterial aggregates seen within the CF lung [50].

Within the context of *P. aeruginosa* infection, a randomized, double-blinded, placebo-controlled trial of a topical phage preparation in patients with otitis media refractory to antibiotic treatment demonstrated both safety and efficacy [51]. *In vitro* work has similarly demonstrated the efficacy of bacteriophages in reducing the colony count of *P. aeruginosa* in sputum samples from CF patients [52]. In animal models of *P. aeruginosa* lung infection (both planktonic and biofilm), the administration of lytic phage preparations lowered both bacterial load and inflammatory markers in the airway [53,54]. Although there have been reports of the use of lytic phages in antibiotic refractory infections [55,56], there are no published clinical trials in humans demonstrating the efficacy of this therapy within the context of CF lung infection. There are, however, trials underway, including a phase 1a/2b trial exploring the safety and efficacy of a nebulized preparation of anti-pseudomonal bacteriophages in CF patients with chronic infection (NCT04596319) [57].

## 3. Strategy: Targeting the Biofilm Matrix

The components of the extracellular matrix are potential therapeutic targets to free bacteria and expose them to both host immune cells, as well as antibiotics. Although there are many components of the extracellular matrix of biofilms, this paragraph will focus on those targeted by drugs with promising results thus far.

### 3.1. Alginate

One example of a biofilm matrix target is the *P. aeruginosa* exopolysaccharide, alginate. *P. aeruginosa* can express a mucoid phenotype through copious production of alginate, which encases bacterial communities, providing a barrier to antimicrobials and host defences [58,59]. Alginate is one exopolysaccharide comprising the extracellular matrix of the biofilm, and is overexpressed in many—but not all—chronic isolates of *P. aeruginosa* from airways of patients with CF. Such a phenotypic change is an example of a survival mechanism that can be driven by factors inherent in the CF lung [58,60], but also by external factors, including antibiotic administration [61].

In the context of CF, there are a number of compounds designed to break through this protective layer. Many have examined the therapeutic potential of degrading enzyme alginate lyase to break through the sticky matrix of polysaccharides. There are several *in vitro* studies demonstrating the efficacy of this compound at disrupting the architecture of the biofilm and increasing susceptibility to antibiotic therapy [62]. More recently, alginate lyase has been combined with antibiotics to degrade biofilms in order to improve penetration of antimicrobials deep into the biofilm. The fluoroquinolone ciprofloxacin, for example, has been formulated for co-administration with alginate lyase in several novel preparations [63,64]. In one, chitosan nanoparticles co-immobilized with ciprofloxacin and alginate lyase effectively reduced the biomass and cell density of biofilm grown *P. aeruginosa* [64].

Alginate itself can be hydrolyzed into low molecular weight (mW]) oligomers that have been used in different biological applications. OligoG CF-5/20 (OligoG) is one such small (mW 2600 Da) alginate oligosaccharide derived from seaweed that has been shown to alter the structure of mucin and increase pore size in the mucus layer, increasing particle diffusion and modifying the viscoelasticity of sputum in CF patients [65]. Early *in vitro* work showed that OligoG can potentiate the action of antibiotics against multi-drug resistant (MDR) Gram-negative bacteria (including *Pseudomonas*, *Acinetobacter*, and *Burkholderia* spp.) as well as in fungal species (including *Aspergillus* and *Candida* spp.) [66,67]. Further work confirmed that OligoG was able to inhibit biofilm formation of Gram-negative organisms [68] and disrupt established biofilm matrix in vitro [69]. OligoG demonstrated an ability to inhibit biofilm formation in a mouse model of *P. aeruginosa* infection in a dose-dependent manner [70]. A randomized, double-blind, placebo-controlled, crossover study investigating a dry powder formulation of OligoG in 65 CF patients with chronic *P. aeruginosa* infection demonstrated safety, and a trend towards an increase in lung function, although results did not reach significance [71]. This approach is still actively being studied, with a dose finding study of OligoG underway (NCT03698448) [72].

### 3.2. Other Exopolysaccharides

In addition to alginate, *P. aeruginosa* produces two other exopolysaccharides: Psl and Pel. Psl and Pel are key components in the formation and maintenance of the biofilm structural framework that provide physical protection to bacteria [73,74]. There is promising work investigating the biofilm dispersion potential of glycoside hydrolases in targeting and degrading Psl and Pel, thereby liberating bacteria trapped within the biofilm and exposing cells to antimicrobial therapy [73]. In a porcine wound model of chronic *P. aeruginosa* infection, PslG hydrolase co-administered with antibiotics demonstrated increased antibiotic penetration into the biofilm as well as improved immune-mediated clearance, when compared to monotherapy [74].

Poly-b-1,6-N-acetyl-D-glucosamine (PNAG/PIA) is another exopolysaccharide important in the structural integrity of biofilms in both Gram-positive and Gram-negative organisms. A hydrolase targeting PNAG called DispersinB showed therapeutic potential in the biofilm dispersal of several different bacterial species (*S. aureus*, *S. epidermidis*, *Enterococci*, *A. baumannii*, *Klebsiella pneumoniae*, and *P. aeruginosa*) in laboratory models of biofilm infection [75].

### 3.3. Extracellular DNA (eDNA)

Extracellular DNA (eDNA) is a key component of the biofilm matrix. Recombinant human deoxyribonuclease I (dornase alfa or DNase) cleaves DNA, released from neutrophils as well as bacteria, that provide a structure to the biofilm. DNase therapy has been shown in meta-analyses to be effective both at reducing the rate of lung function decline, as well as reducing the number of pulmonary exacerbations [76]. DNase also disrupts pre-formed bacterial biofilms, which may account for some of its effect [77]. Although the predominant mechanism has been attributed to the reduction in mucous viscosity and improved airway clearance, DNase also disperses bacteria and increases antimicrobial susceptibility in staphylococcal species [77] as well as in mixed bacterial species models of biofilm infection [78]. Furthermore, the discovery of bacterial expressed eDNA binding proteins that crosslink with eDNA strands within the matrix provides potential new therapeutic targets for the eradication of biofilms [79]. As a proof of concept, it has been shown that *Burkholderia cenocepacia* biofilms were dispersed by targeting the DNA binding integration host factor (IHF) but could not be dispersed by DNase treatment [80].

## 4. Strategy: Targeting Intracellular Signaling Pathways

Bis-(3′-5′)-cyclic dimeric GMP (c-di-GMP) is the principle second messenger involved in the regulation of genes that are expressed in a biofilm lifestyle [81]. Altering intracellular signalling mechanisms has the potential to catalyze the transition of bacteria from a biofilm lifestyle into a planktonic one, thus rendering the dispersed bacteria more susceptible to antimicrobials. There are many signalling pathways with intracellular mediators that have been implicated in biofilm formation and bacterial virulence, but here we focus on the pathways that have yielded the most promise in terms of pharmaceutical intervention.

### 4.1. Compounds Reducing Intracellular Cyclic-di-GMP

Increased levels of c-di-GMP promote transition from planktonic to a biofilm mode of growth, whereas reduction in levels results in biofilm dispersal [82]. Cells are disseminated from *P. aeruginosa* biofilms through the modulation of specific phosphodiesterases, which are activated in response to environmental cues. Blocking or downregulating this signalling pathway has therefore been proposed as a promising drug target to aid in the eradication of biofilms.

Previous studies demonstrated that a decrease in c-di-GMP level, led to the dispersal of established *P. aeruginosa* biofilms on infected silicone implants in the peritoneal cavity of mice [83]. In the treated mice, bacteria accumulated in the spleen, and although no mice died or experienced sepsis, this finding would suggest adjunctive use of antimicrobials to minimize the risk of widespread bacterial dispersal. Different groups have since used high throughput methods to detect compounds that reduce the levels of c-di-GMP in a number of different bacterial species [84]. Azathioprine, an immunosuppressive drug already in clinical use, is a compound shown to decrease di-GMP, correlating with a lack of expression of extracellular matrix proteins in *E. coli* [85].

### 4.2. Other Compounds Reducing Cyclic-di-GMP: Nitric Oxide (NO)

Nitric oxide (NO), while long recognised as a vital signalling compound in eukaryote biology [86], has now been implicated as one of the key intracellular signalling molecules involved in bacterial growth—namely, in the transition between planktonic and biofilm lifestyles. Highly water soluble and diffusible in biological systems, NO is ideal for intracellular communication. Although high levels of NO can be toxic to bacteria, some bacterial species encounter high levels of this compound in denitrification, a process whereby certain prokaryotes can use nitrate or nitrate for respiration in oxygen-limited conditions [87]. Exogenous NO has been shown to improve antibiotic susceptibility in resistant bacteria and has therefore been suggested as an adjuvant therapy [88]. It is believed that NO works by increasing concentrations of specific PDEs which, in turn, reduce the levels of c-di-GMP and disperse cells from the biofilm [89,90,91]. Treatment of *P. aeruginosa* biofilms with NO has been shown to result in biofilm dispersal [89,90], but not kill planktonically grown cells [92]. There are currently three approaches to using NO therapeutically in bacterial infection: Delivering NO donor prodrugs, direct inhalation of NO, and increasing endogenous NO formation.

NO donor prodrugs are designed to release NO locally, thereby triggering biofilm dispersion. Screening of chemical libraries first led to the identification of sulfathiazole [93], an antimetabolite drug, as a potential compound targeting this pathway. More recently, cephalosporin-3′-diazeniumdiolates (C3Ds) have been shown to release NO following cleavage of their β-lactam moiety with bacterial β-lactamases. *In vitro* work demonstrated that this compound caused significant *P. aeruginosa* biofilm dispersal on its own, improved dispersal when co-administered with tobramycin, and nearly completely eradicated biofilms when combined with colistin [94]. The donor NO compound spermine NONOate (S150), has also shown efficacy in reducing biofilm biomass of *P. aeruginosa* clinical isolates from CF patients [92].

Inhaled gaseous NO has also been studied; results from a small pilot study in CF patients with *P. aeruginosa* infection yielded lower numbers of bacterial aggregates in the sputum when compared with controls when used as adjunctive therapy to antibiotics [95]. A clinical trial of high dose (160 ppb) inhaled NO in CF patients is currently underway (NCT02498535) in the USA [96].

Another approach is to increase endogenous NO production. One way of doing this is through inhibition of arginase, an enzyme that competes with nitric oxide synthase (NOS) for L-arginine as substrate. A phase 1B randomized, double-blind, placebo-controlled trial investigating a compound called CB-280, a potent arginase inhibitor, is currently being conducted (NCT04279769) [97].

## 5. Strategy: Targeting Cell-Cell Signalling

Quorum sensing (QS) is a method by which bacteria can communicate with one another through the production, secretion, and sensing of small molecules called autoinducers, which allows for regulation of population level phenotypic change [98]. QS allows for a bacterium to modulate gene expression in response to gradients of bacterial density, directing lifestyle choices appropriate to a given environment. Many cellular processes, including but not limited to, the transition to a biofilm mode of growth and virulence factor expression, are known to be controlled by the QS system [99]. As the primary method of communication between bacteria, QS has also been implicated as a key factor in the intracellular pathways mentioned previously. It has been shown to regulate levels of c-di-GMP, the principle second messenger involved in the regulation of genes that are expressed in a biofilm lifestyle [81]. Bacteria transitioning from a biofilm lifestyle into a dispersion phase, change phenotypically [100], enabling it to move to new areas and colonize different regions of the lung.

In many Gram-negative bacteria, QS is controlled by two N-acyl-homoserine lactone (AHL) signalling systems—the las and rhl systems [101,102]. Whereas the las system is activated in cells undergoing early biofilm formation [103], RhlR/RhlI is activated in the maturation stages of biofilm formation [100]. These systems are hierarchically arranged, whereby the las system has the power to up-regulate or downregulate rhl [104]. In Gram-positive bacteria, autoinducing peptides (AIPs) fulfill the same role, whereas both Gram-positive and negative organisms produce autoinducer-2 (AI-2) [105]. These QS pathways are ubiquitous in bacterial cells and have specific roles in biofilm development and maintenance.

### 5.1. Compounds That Disrupt Quorum Sensing

The seaweed Delisea pulchra was observed to avoid bacterial colonization by the production of halogenated furanone compounds similar in structure to the homoserine lactones used in QS in Gram-negative organisms [106]. These findings led to the concept of using such compounds therapeutically. Kim et al. [107] synthesized furanone derivatives that were noted to inhibit QS and impair biofilm formation, suggesting that compounds inhibiting QS may aid in bacterial eradication. Subsequently, several more compounds were identified through high throughput screening methods, detected through suppression of genes normally expressed in QS [108]. Compounds have also been identified by computerized structural analysis models [109]. One promising compound that inhibited QS was garlic extract—in particular the compound ajoene [110,111]. In a mouse model of pulmonary *P. aeruginosa* biofilm infection, ajoene was effective in blocking the production of rhamnolipids needed for biofilm maintenance [108]. However, a small, randomized pilot study exploring the effects of oral garlic in CF patients with chronic *P. aeruginosa* infection showed no significant effect on pulmonary function [112].

Several antibiotics already in use have effects on QS. For example, the macrolide azithromycin, whilst not bactericidal against *P. aeruginosa*, can disrupt quorum sensing and block alginate production in a mouse model of infection with this organism [113]. In a meta-analysis including four trials of CF patients, those treated with azithromycin had improved lung function, especially in the subgroup of patients colonized with *P. aeruginosa* [114]. It is worth noting that there are other hypotheses regarding the clinical efficacity of azithromycin, including effects on airway inflammation [115,116]. Furthermore, the antibiotics ceftazidime and ciprofloxacin have also been shown to disrupt QS [117]. On this basis, identification of compounds using this mechanism for targeting biofilm bacteria remains one that holds promise.

### 5.2. Compounds Disrupting Iron Metabolism

Successful pathogenic bacteria often employ the use of iron scavenging systems to obtain iron, an essential nutrient for bacterial growth, from host tissues [118]. Iron is also an important signalling molecule involved in bacterial adherence and biofilm maturation in certain bacterial pathogens. Previous studies demonstrated that iron depletion in models of *P. aeruginosa* and *B. cenocepacia* infection led to widespread transition from biofilm to a motile planktonic mode of growth, while increasing the availability of iron led to bacterial aggregation [119].

Gallium (Ga^3+^) is a metal that interferes with iron signalling. Ga^3+^ and Fe^3+^ have many similarities in size and other electrochemical properties that allow gallium to be mistaken for iron and taken up by bacterial cells, subsequently disrupting cellular processes promoting biofilm formation [120,121]. This approach has been shown to be efficacious in murine models of *P. aeruginosa* infection—in two models of lung infection [122] and in a thermally induced wound infection [123]—while simultaneously associated with lower rates of bacterial resistance compared to other small molecule antibiotic therapies [120]. In an experimental model using lung epithelial cells, a liposomal formulation of gentamicin co-encapsulated with gallium eradicated *P. aeruginosa* biofilms more efficaciously than liposomal aminoglycoside alone [124]. In human clinical trials, results from a recent Phase 2 study in 23 centres in the U.S. demonstrated trends towards improved lung function and lower *P. aeruginosa* density in sputum in CF patients with chronic *P. aeruginosa* infection treated with intravenous (IV) gallium [125]. There are also clinical trials in development to evaluate IV gallium in the treatment of NTM pulmonary infection (NCT04294043) [126].

## 6. Conclusions

Effectively treating bacterial infections in the context of CF entails a thorough examination of the environments in which these pathogens survive despite repeated, broad spectrum courses of antibiotics. The biofilm is an adaptive structure that provides bacterial populations with both physical protection and reservoirs of phenotypically distinct subpopulations that have the ability to withstand antimicrobials and immune response. Though existing therapies have yielded improved patient outcomes, including lung function, survival and quality of life, the biofilm lifestyle represents a barrier that limits benefit. There are a number of therapies currently in use that have anti-biofilm properties beyond their initially understood mechanism of action—from the eDNA lysing dornase alfa to the anti-quorum sensing properties of azithromycin. Such therapies, when used in conjunction with standard antimicrobials, serve as examples of using multiple modes of action against biofilms to improve clinical outcomes.

It is also important to consider what impact improved CFTR function will have on microbial communities, in an era where an increasing number of people with CF have access to effective CFTR modulator therapy. Though we do not yet have the answer, the multifactorial nature of the question has been the focus of several reviews [127,128]. However effective these new therapies prove to be, the need for anti-biofilm agents will continue to be relevant in the context of CF. The biofilm lifestyle is also but one of many factors that need to be taken into account when designing laboratory methods to more accurately predict response to treatment, as studies investigating antimicrobial selection based on biofilm versus standard AST in the treatment of CF patients have not yielded improved clinical outcomes [129,130]. It may be that the next breakthrough in targeting microbial biofilms lies in a personalized approach to bacterial eradication, accounting for the influence of polymicrobial infection or other patient-specific characteristics in models of disease. 

## Figures and Tables

**Figure 1 ijms-22-02155-f001:**
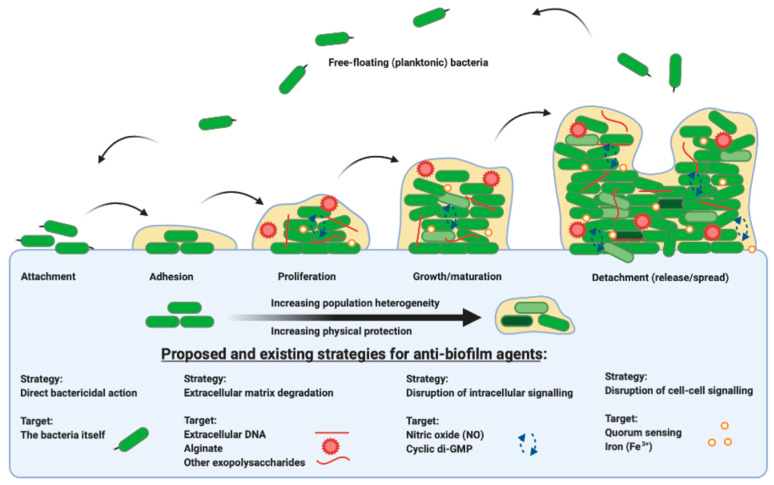
A representation of the bacterial biofilm life cycle, from the point at which free-floating planktonic cells attach to a substrate or surface, to adhesion, to proliferation and microcolony formation, through to growth and maturation and subsequent dispersal, propagating and beginning the life cycle once more. There are many existing and proposed strategies for targeting biofilm grown organisms, from direct action on the bacteria themselves, to the extracellular matrix, to intracellular signaling pathways, to the disruption of cell-to-cell signaling. Figure created in BioRender.com.

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
