# Peer review of "Approaches to Targeting Bacterial Biofilms in Cystic Fibrosis Airways"

_ijms, 2021, doi:10.3390/ijms22042155_

Round 1

Reviewer 1 Report

This article is highly organized about the strategies against biofilm-related lung infection diseases. I consider it provides information for many readers including biofilm researchers, pharmaceutical researchers, medical doctors, and so on.  I would like you to confirm and deal with below some points before re-submit the article.

1) Please change the following names in Italic.

L105: Acinetobacter baumani

L129: S. aureus

L254: E. coli

L275: Delisea pulchra

2) L166-178: Is Oligo G same as Oligo G CF-5/20?

3) L219: There are two “nitrate”, I think one of them is other compound.

4) L263-4: I think full-name of Bis-[3’-5’]-cyclic dimeric GMP, and the part of “the principle second messenger involved in the regulation of genes that” are not necessary. Because the information has been provided in a previous paragraph.

Author Response

Dear Reviewer,

Many thanks for your feedback on our manuscript. Attached, you will find an updated manuscript reflecting your suggested changes, as well as those of the other reviewers. 

I will go through your points individually to show you how the manuscript reflects these changes:

  • 1) Please change the following names in Italic: L105: Acinetobacter baumani, L129: S. aureus, L254: E. coli, L275: Delisea pulchra

All of these changes have been made, along with some others in the references section that had been missed. 

  • L166-178: Is Oligo G same as Oligo G CF-5/20?

Yes. We have changed the manuscript to reflect this change.

  • L219: There are two “nitrate”, I think one of them is other compound.

Yes, the manuscript has been changed to reflect this. 

  • L263-4: I think full-name of Bis-[3’-5’]-cyclic dimeric GMP, and the part of “the principle second messenger involved in the regulation of genes that” are not necessary. Because the information has been provided in a previous paragraph.

This change has also been made to the manuscript. 

Many thanks for your help in making this manuscript ready for publication. We are very grateful for your comments.

All the best,

Isaac Martin (on behalf of Val Waters and Hartmut Grasemann)

Reviewer 2 Report

  • Please, check the typography (italic style, space), even in the Reference part.
  • Lines 58-61 : Can you give more precisions about persistent cells ? How do they appear ? Is it an automatic process ?
  • Lines 62-63 : Can you specify recent "official" data about the involvement of biofilms in chronic diseases ?
  • Figure 1 : it would be more appropriate to display the figure before the paragraph beginning by "We now understand..." (lines 62-67).
  • The "Proposed and existing strategies for anti-biofilm agents" part of the Figure 1 is not clear. It shows the targets of the potentiel anti-biofilm agents but also antimicrobial components (antibiotics or phages) and it is indicated "mechanism". Please clarify.
  • Lines 82-83 : Are you sure to refer to MBEC rather than MBIC ? Can you give a reference ?
  • Lines 98-99 : Can you specify the most frequent molecules used for CF treatments ?
  • "Bacteriophage" paragraph : some "s" are missing, please verify.
  • Line 160-161 : Redundancy of "alginate lyase", please rephrase.
  • Line 95 : Can you precise the role and the origin of eDNA ?
  • Part 4 : At first sight, we don't understand the link between the NO paragraph and the Part 4 introduction, please rectify (or replace the "Compounds reducing intracellular c-di-GMP" paragraph before ?).

Author Response

Dear Reviewer,

Many thanks for your feedback on our manuscript. Attached, you will find an updated manuscript reflecting your suggested changes. 

I will go through your points individually to show you how the manuscript reflects these changes:

  • Please, check the typography (italic style, space), even in the Reference part. 

There were many instances throughout the manuscript where the names of bacterial species had not been italicised. The same was true of the reference section. These mistakes have been corrected. 

  • Lines 58-61 : Can you give more precisions about persistent cells ? How do they appear ? Is it an automatic process ?

There is now an extra line in the manuscript about how it is thought that persister cells arise in cases of nutrient scarcity and in the presence of noxious substances as part of a survival process. 

  • Lines 62-63 : Can you specify recent "official" data about the involvement of biofilms in chronic diseases?

We have touched upon the role of biofilms in other chronic diseases (lines 70-74), but going into depth about how biofilm infection differs in these different diseases is beyond the scope of this review. 

  • Figure 1 : it would be more appropriate to display the figure before the paragraph beginning by "We now understand..." (lines 62-67).

This suggestion was implemented. 

  • The "Proposed and existing strategies for anti-biofilm agents" part of the Figure 1 is not clear. It shows the targets of the potentiel anti-biofilm agents but also antimicrobial components (antibiotics or phages) and it is indicated "mechanism". Please clarify.

This was a very good point. The figure has now been changed to reflect this error. 

  • Lines 82-83 : Are you sure to refer to MBEC rather than MBIC ?

This was a good point. The reference looked at the biofilm inhibitory concentrations rather than the MBEC. The manuscript has been changed to reflect this. 

  • Lines 98-99 : Can you specify the most frequent molecules used for CF treatments ?

We have mentioned a few of the heavy hitters in the manuscript (colisin, tobramycin); however, these agents change depending on the bacterial species and clinical response. 

  • "Bacteriophage" paragraph : some "s" are missing, please verify.

There is no consensus on whether bacteriophage or phage should be pluralised with an S. For what it's worth, there is even some disagreement amongst the authors of this paper! We have, however, opted to pluralise bacteriophage with an (s). 

  • Line 160-161 : Redundancy of "alginate lyase", please rephrase.

Changed.

  • Line 95 : Can you precise the role and the origin of eDNA ?

We have added a line about how eDNA results from both the lysis of different bacterial species, but also neutrophils. 

  • Part 4 : At first sight, we don't understand the link between the NO paragraph and the Part 4 introduction, please rectify (or replace the "Compounds reducing intracellular c-di-GMP" paragraph before ?).

This was well pointed out. We have changed the order of the sections so that it reads better. 

Many thanks for your help in making this manuscript ready for publication. We are very grateful for your comments.

All the best,

Isaac Martin (on behalf of Val Waters and Hartmut Grasemann)

Reviewer 3 Report

The manuscript by Martin et al., is an extensive review of anti-biofilm strategies approved for clinical use or under experimental investigation for the treatment of Cystic Fibrosis. This is a remarkably well-written review and is interesting.

Specific comments:

  • Anti microbial peptides are an emerging therapeutic approached targeting bacterial biofilms for CF. Therefore, this reviewer suggest to include a paragraph about antimicrobial peptides such as: LL-37, Colistin, 6K-F17. It has been demonstrated by different groups that these antimicrobial peptides are highly effective in disrupting and killing pre-formed multidrug resistant P. Aeruginosa biofilms and reduce the LPS-induced pro-inflammatory responses  (doi: 10.4049/jimmunol.176.4.2455; doi: 10.1038/s41598-018-33016-7; doi: 10.3390/biom10020334; doi: 10.1002/psc.2674). Moreover, recently Deber’s lab demonstrated that the peptide 6K-F17 restore Orkmabi rescue of F508del-CFTR function in airway cells infected with clinical strains of P. aeruginosa (doi: 10.3390/biom10020334). The combination of CFTR modulators and antimicrobial peptides could be a therapeutical approach to improve the efficacy of CFTR modulators in CF patients. Since the authors mentioned the impact of CFTR modulators on bacterial infection in the conclusion section, a short discussion about the effect of infection on CFTR modulators will be helpful for the reader.
  • Page 6, lanes 237-239; it has been demonstrated that inhibition of arginase by the compound CB-1158, currently in clinical trial (NCT02903914) increased cytosolic NO and enhanced the rescue effect of ORKAMBI on F508del-CFTR-mediated function in primary bronchial and nasal cells. (doi: 10.1124/mol.119.117143)
  • Page 7, “Compounds that disrupt quorum sensing”. The authors may want to discuss the paper by Maillè et al. where it has been demonstrated that treatment of P. aeruginosa cultures with a quorum sensing inhibitor (HDMF) prevented the negative effect of P. aeruginosa exoproducts on Wt-CFTR and preserved CFTR rescue by CFTR modulators in CF airway epithelial cells.

Author Response

Dear Reviewer,

Many thanks for your feedback on our manuscript. Out of all the reviewers' comments, we spent the most time discussing your ideas, which were very thought provoking and informative. Attached, you will find an updated manuscript highlighting yours, as well as the other reviewers' changes that have been implemented. 

Below, I will highlight your points individually to explain how we have addressed your feedback. In instances where we have opted not to change the manuscript, we have tried to justify our reason for doing so. 

  • Antimicrobial peptides

There is now a section on AMPs, using some of the references that you helpfully provided. 

  • CB-1158

We opted not to include this compound that was designed and tested as a cancer drug as it was chemically altered to CB-280 for specific use in CF patients, the compound discussed in this review. The studies with CB-1158 form the scientific basis for the compound discussed in this review. 

  • The study by Maillè et al. with QS inhibitors and CFTR modulators

We opted not to include this very interesting study due to the fact that it was (1) not specific for biofilm infection and (2) it seemed a little beyond the scope of this review. We have indicated several reviews in the manuscript that deal with this sort of question should the reader wish to pursue it. 

Many thanks for your feedback on our manuscript. Your comments have made this a stronger article and we are very grateful for your time. 

Sincerely,

Isaac Martin (on behalf of Val Waters and Hartmut Grasemann)